# AMH: Could It Be Used as A Biomarker for Fertility and Superovulation in Domestic Animals?

**DOI:** 10.3390/genes10121009

**Published:** 2019-12-04

**Authors:** Saqib Umer, Shan Jiang Zhao, Abdul Sammad, Bahlibi Weldegebriall Sahlu, YunWei Pang, Huabin Zhu

**Affiliations:** 1Embryo Biotechnology and Reproduction Laboratory, Institute of Animal Sciences, Chinese Academy of Agricultural Sciences, Beijing 100193, China; saqibumar33@hotmail.com (S.U.); zhaoshanjiang@caas.cn (S.J.Z.); blenbah@gmail.com (B.W.S.); pangyunwei@caas.cn (Y.P.); 2Key Laboratory of Animal Genetics, Breeding and Reproduction, CAST, China Agricultural University, Beijing 100193, China; drabdulsammad1742@yahoo.com

**Keywords:** AMH, AFC, superovulation, fertility, domestic animals

## Abstract

Anti-Müllerian hormone (AMH) is a reliable and easily detectable reproductive marker for the fertility competence of many farm animal species. AMH is also a good predictor of superovulation in cattle, sheep, and mares. In this review, we have summarized the recent findings related to AMH and its predictive reliability related to fertility and superovulation in domestic animals, especially in cattle. We focused on: (1) the dynamics of AMH level from infancy to prepubescence as well as during puberty and adulthood; (2) AMH as a predictor of fertility; (3) the association between antral follicle count (AFC) and plasma AMH level; (4) AMH as a predictor of superovulation; and (5) factors affecting AMH levels in domestic animals, especially cattle. Many factors affect the circulatory levels of AMH when considering the plasma, like nutrition, activity of granulosa cells, disease state and endocrine disruptions during fetal life. Briefly, we concluded that AMH concentrations are static within individuals, and collection of a single dose of blood has become more popular in the field of assisted reproductive technologies (ART). It may act as a potential predictor of fertility, superovulation, and ovarian disorders in domestic animals. However, due to the limited research in domestic animals, this potential of AMH remains underutilized.

## 1. Introduction

Anti-Müllerian hormone (AMH) has a long history, but its presence was recognized after the mid-20th century. Alfred Jost was the first to introduce the existence of AMH in 1953. At that time, scientists thought that testicular tissues not only synthesize testosterone, the chemical messenger responsible for the development of male external genitalia, but also produce a chemical that regressed Müllerian ducts in rabbit fetuses [1]. AMH was later characterized by Picon [2] and then purified from the incubation media of bovine fetal testicular tissue [3]. In late fetal life, AMH is secreted by ovarian granulosa cells of females (in women [4], cattle [5], and sheep [6]), when Müllerian ducts are no longer responsive to the hormone [7,8]. AMH is a member of the transforming growth factor beta (TGF-β) family and is also called Müllerian inhibiting substance/factor (MIS) [9]. It is a glycoprotein in nature, with a molecular weight of 140 kDa corresponding to 553–575 amino acids [10] and a half-life of 1.5 days [11]. The *AMH* gene has been mapped to chromosome 7 in cattle, mares, and goats; chromosome 5 in sheep; chromosome 9 in buffalo; and chromosome 2 in pig [12,13]. Recently, AMH has become a potential reproductive biomarker for predicting the ovarian pool of follicles in donor cows [14]. Hence, this review mainly highlights the importance of AMH as a fertility and superovulation biomarker in domestic animal species, especially in cattle.

## 2. Anti-Müllerian Hormone Signaling Pathways

The superfamily TGF-β has over 30 ligands, including bone morphogenetic proteins (BMPs), which are the largest subfamily, as well as growth and differentiation factors (GDFs) [15,16]. Previously, AMH was considered as an indirect member of the TGF-β superfamily but due to the analogy with the signaling mechanism of BMPs, however, it is now considered a direct member of this family [17]. The TGF-β family members act through two types of heteromeric receptors (type I and type II), which further consist of two subtypes, i.e., serine and threonine. Sometimes, the co-receptors beta glycan and endoglin also help during signaling [15,16]. After ligand binding results, type II receptor-mediated phosphorylation takes place, activating the type I receptor that ultimately leads to the activation of several pathways, e.g., Smad, mitogen-activated protein kinases (MAPK), and phosphatidylinositol 3-kinase (PI3K)/Akt. Through the activation of Smad4, the AMH target gene regulates transcription [15,16,18,19]. The schematic mechanism of AMH signaling is illustrated in Figure 1.

In mammals, five different type II receptors have already been identified, with AMHRII specifically involved in AMH signaling [17,20] while three BMPs (ActRII, ActRIIB, and BMPRII) were found to be involved in other signaling pathways [21]. Similarly, seven subtypes of type I receptors have also been identified in mammals (anaplastic lymphoma kinase (ALK)1–7) [21]. Among these, ALK2, ALK3, and ALK6 perform functions related to AMH [17]. Different types of BMPs are produced in different cells of the ovary and each perform their unique respective functions; BMP4 and BMP7 are expressed in theca cells [22,23] while AMH, BMP2, and BMP6 are produced in granulosa cells [24]. In goats, BMP15 regulates AMH by triggering the MAPK pathway [25]. However, owing to the fact that the signaling pathways perform complex and distinct functions in the ovary, detailed studies are required for a better understanding of ligand and receptor expression as well as the interaction and communication of binding proteins with the surrounding cells.

## 3. Role of AMH

AMH production starts as early as the initial selection of ovarian follicular waves [26]. AMH expression reaches its peak level in primordial, primary, and secondary follicles, whereas it decreases once the dominant follicle is selected and is absent in atretic follicles. This dynamic expression was firstly reported in rabbits (rodents) [27], and then in women [28], cattle [29], sheep [30], buffalo [31], goats [32], mares [33], and pigs [34]. Following ovulation in pigs, AMH expression then continues in the luteal cells of corpus luteum. The exact physiological function in luteal cells remains unknown, but it is speculated that it might be to regulate the cyclic recruitment of small antral follicles by avoiding premature exhaustion of the ovarian follicular reserves [34]. AMH controls the number of follicles and selection of the dominant follicle during follicular waves. The recruitment of follicles is faster in the absence of AMH, but the ovarian follicular reserves exhaust at a younger age [35]. AMH has been identified to suppress follicle-stimulating hormone (FSH) receptors in gonadotropin-dependent small antral follicles [36]. In growing follicles, the FSH sensitivity of granulosa cells was reduced in the presence of AMH [37]. In the presence of luteinizing hormone (LH), thecal cells produce androgens and are transformed into estrogen by the aromatase system of granulosa cells. This process of theca and granulosa cells happens under the influence of FSH. Growing follicles with theca cells have the potential to synthesize androgens, but few of them possess the aromatase system in granulosa cells. Therefore, the possession of the aromatase system is under the control of FSH in granulosa cells with more FSH receptors, leading to a high proliferation of granulosa cells which ultimately selects and precedes the dominant follicle [38]. In another study, De Clemente et al. proposed a regulatory role for AMH on follicular development and maturation. AMH retards the expression of the aromatase enzyme and suppression of the LH receptor on the surface of granulosa cells [39], representing a renowned phenomenon in which meiotic division is arrested at prophase I, during oogenesis, and is physiologically resumed after females have undergone puberty [40]. In rat oocytes, germinal vesicle breakdown (GVBD) was inhibited by AMH during in vitro experiments [41]. Furthermore, both in vivo and in vitro experiments in mice have shown that AMH also inhibits FSH superstimulated follicular growth [36]. As shown in Figure 2, AMH plays two critical functional roles in females: 1) it inhibits primordial follicular growth from the ovarian follicular pool by avoiding premature exhaustion of the follicular reserves of the female, and 2) it reduces the sensitivity of the preantral and small antral follicles to FSH while modulating follicular development [42].

## 4. How Antral Follicle Count Became a Fertility Biomarker

In the bovine estrus cycle, the growth of follicles occurs under the influence of FSH at an interval of 7–10 days [43]. The growing follicles (≥3 mm in diameter) were monitored by ovarian ultrasonography which revealed that the per wave antral follicle count (AFC) was highly variable (ranging from 8–54 follicles) but highly repeatable (1 = perfect, 0.95) during the same and the subsequent estrus cycle within each animal [44]. This study opened a new portal for confirmatory research. Ireland confirmed these findings with data from a large number of animals (n = 69) by monitoring 188 follicle waves [45]. In young adult beef cattle, the size of the ovary was positively associated (*r* = 0.89, *p* < 0.001) with low vs. high AFC [46]. The basal FSH concentrations were negatively correlated with the number of follicles in dairy [47] and beef [45] heifers, and non-lactating dairy [48] and lactating beef cows [49]. The concentration of progesterone and thickening of the endometrium remains constant with both low and high AFC (*p* < 0.01) [50], while they affect the embryo mortality in cattle [51] and cause infertility in women [52]. The ovarian androgen production has no effect on low vs. high AFC in cattle [53]. In conclusion, cattle can be a phenotypically reliable animal for AFC because repeatability was not affected by age, breed, season, stage of lactation, level of hormone, nor time span of the AFC measurement [44,45,46,47,50,54,55].

## 5. AMH Repeatability and Relationship with AFC

Emerging knowledge depicts that AMH level slightly varies during the estrus cycle of cattle. The measurement of the AMH level in young adult beef heifers at a single time point was found to be highly positively associated (*r* = 0.97) with the average of the AMH level determined from measurements at multiple time points at different days of multiple estrus cycles [56,57]. The AMH level remained static during the estrus cycle of dairy cows [56,58,59] on different days of two estrus cycles [58], and also during the natural and synchronized estrus cycles within the same individual [60]. The average circulating level of AMH for every single cow was positively correlated (*r* = 0.65; *p* < 0.01) with a superstimulated response (number of corpus luteum at the time of flushing) and total collected embryos (*r* = 0.50; *p* < 0.01; [59]. A positive correlation was observed between the plasma level of AMH and the numbers of ova/embryos, fertilized embryos, and transferable embryos in the Japanese black cow, which suggested that these AMH concentrations are useful for predicting early-stage markers for selecting Japanese black donor cows [61,62]. In mares, AMH showed a positive association with AFC within and between estrous cycles, i.e., middle-aged (9–18 years) and old mares (19–27 years) [63]. In Barki sheep, circulating AMH level was positively correlated with antral follicles (*r* = 0.88) and progesterone (P_4;_
*r* = 0.41) [64]. The above findings indicate that AMH concentration could be a reliable biomarker based on a single random blood sampling at any day of a cycle in adult cattle mare and sheep.

AMH was highly positively correlated (*r* = 0.90) with the disparity in AFC and the histologically determined total number of morphologically healthy follicles (primordial, transitory, primary, secondary, and antral) and oocytes in the ovaries of young adult cattle [46]. The overall mean AMH amount during ovulatory follicular waves per animal had a highly significant correlation (*r* = 0.92) with an average peak AFC during two or three waves of an estrous cycle [46]. AMH repeatability was high between post-weaning and pre-service evaluations, which indicated that post-weaning maximum AFC and AMH concentrations may be applied to select Bradford heifers that start puberty at an early age [65]. A significant positive correlation was also assessed between AMH and the number of follicles in dairy Holstein, European, and Zebu cattle [58,66,67]. In mares, the repeatability of AFC and plasma AMH level was high because AMH was consistent within and between estrus cycles [63]. In sheep, AMH repeatability was low within-animal because it reached its peak concentration at different times among different animals [68]. In individuals with either a low or high AFC, this value did not show any significant association with AMH [69]. These facts highlight the importance of the relationship and reliability of both AMH and AFC as predictive reproductive biomarkers for the size of the reserve ovarian pool in age-matched cattle and mares, but it appears limited in sheep, and so more reports are required based on data derived at a larger scale.

## 6. Fertility and Dynamics of AMH in Different Age Groups

An experiment on Maine-Anjou beef heifers illustrates the variable level of AMH. In calves from one to three months of age, the level of AMH was increased until six months of age, and then slowly decreased from seven to 12 months. The age of first ovulation was one year for this breed [70]. Similar dynamics were found in female Holstein calves, in which the level of AMH was increased until two months of age, started to decline in the fifth month, and then stabilized when the ovulation age reached eight to nine months [70]. The AMH concentration in mares increases at an early age and reaches its peak level at around 16–18 years, and the level of AMH then declines with increasing age [71,72]. In sheep (Rasa Aragonesa; bred for wool and meat), the AMH level was not associated with lambs between prepubescence and adulthood. In prepubescent lambs, the AMH level increases from 3–4.5 months of age, while it decreases in the sixth month [68]. In another study of sheep (Sarda; bred for dairy purposes), AMH concentration tended to increase until two to five weeks of age and decline at six weeks of age [69]. These studies indicate that the AMH level was high during early life as compared to a young age in all species, but there is variation due to the age of puberty onset in different domestic species [73].

AMH has also been recently proposed as a potential biomarker for precocious puberty and weaning [65,74]. It is well known that heifers with precocious puberty (≤10 months) can be bred at a lower cost than breeds with later puberty [75]. Moreover, precocious puberty allows the heifer to have more estrous cycles before breeding age, increased first-service conception rate [76], earlier pregnancy [77], and enhanced lifetime productivity [78]. Ali et al. sampled Japanese black female calves from their first week after birth until their sixth week of early puberty, and observed characteristic AMH level trends, concluding that higher levels of AMH during this period could decide the early onset of puberty and characteristic post-puberty AMH levels [74]. Later on, another study also indicated that post-weaning AMH levels may be useful for selecting Bradford heifers with precocious puberty [65]. However, these types of studies need to be evaluated at large-herd and multibreed levels to confirm the predictive ability of AMH as a biomarker for precocious puberty. Jimenez-Krassel et al. conducted a study on young adult Holstein heifers (age 11–15 months, n = 281). Heifers were divided into four quartiles based on their circulating mean AMH concentrations (Q1 = 19 pg/mL, Q2 = 41.8 pg/mL, Q3 = 68.9 pg/mL, and Q4 = 153.2 pg/mL); moreover, several parameters of reproductive performance before and after calving were analyzed in every individual before the start of third lactation. Conception and pregnancy rates after the first artificial insemination (heifers averaged 44.5%; n = 240 animals) did not differ among quartiles. A lack of difference in results has opened a new discussion due to unexpected results with different levels of AMH [79]. If we compare low vs. high AFC heifers, low AFC heifers show diminished ovarian function, oocyte quality, and endometrial development [50,54,80]. Another study reported that high AMH dairy cows had higher pregnancy rates and a lower incidence of pregnancy loss between 30 to 65 days of gestation [81]. There are no significant relationships between AMH and AFC at a young age (three to eight years) in mares [63]. From these studies, we conclude that heifers showed suboptimal findings of AMH after the birth of a calf, and further research into heifers and young mares are thus needed to establish AMH as a predictive marker of fertility.

Productive herd life (time in the herd after calving) was positively associated with AMH in heifers. In the division of the above-discussed study, further analysis revealed that Q1 cows (low AMH group) had a short lactation period (180 days) as compared to Q2 and Q3 cows. The percentage of cows in Q1, Q2, Q3, and Q4 were respectively 24%, 37%, 43%, and 32% after culling in the herd. This result indicates that the probability of culling was high in Q1 group as compared to the other three groups (Q2, Q3, Q4). These findings lead to the conclusion that a single test of circulating AMH concentration in young heifers can predict herd longevity [79].

It has been shown that the quantity of AMH varies slightly during the same days of two estrus cycles in the same cattle [58,59]. In dairy cows, during the natural and synchronized estrus cycle of the same individual, the AMH concentration remains the same [60], while in goats, the level of AMH and FSH increased after synchronization as compared to a natural estrus cycle [82]. In beef heifers, the value of a single AMH measurement was strongly correlated with the average of multiple measurements of AMH during different days of multiple estrus cycles [57]. Altogether, these reports summarized the stable nature of AMH during the estrus cycle and its repeatability among multiple estrus cycles in bovines and mares. These studies further highlight the potential utility of AMH as a reproductive marker based on a single blood sample from adult cattle and mares.

## 7. AMH and Assisted Reproductive Technologies

In assisted reproductive technologies (ART), both AMH and AFC were used as markers for superstimulation, but the response of superovulation was negatively correlated with the number of follicles and ova in cattle ovaries [49,83,84,85]. Over time, scientific reports in the favor of AMH as a marker for superovulatory response has increased. AMH had a positive association with follicles before and after treatment, and with the corpus luteum (CL) of the ovary [58]. In dry dairy cows, AMH showed a high correlation with the number of graphian follicles and number of embryos collected using the multiple ovulation and embryo transfer (MOET) protocol [86], and a similar relationship was found in Japanese black beef cattle [62,87]. A superstimulation response was also correlated with circulating AMH levels of lactating Holstein cows [59]. A positive association was established between ovum pick-up (OPU) and AMH in *Bos indicus* (Zebu; [88], beef Korean Hanwoo; [89] and Holstein [90] cows) for MOET. Furthermore, AMH measurements were also used as a predictive marker for the superovulatory response in goats [91], sheep [68], mares [33], and buffalo [31]. In donor ewes, AMH can be used as a reliable marker for superovulation and the in vivo embryo production response [92]. This evidence confirms that the concentration of AMH could be used as predictive marker for the ovulation response in the field of ART, especially in superovulation.

## 8. Heritability of AMH

In dairy and beef cattle, the heritability of female reproductive traits is low [93]. In spite of that, the advancement of genetic improvement by the identification of biomarkers has a positive association with moderate to highly heritable traits and fertility in dairy cattle. Two reports were recently established to estimate AMH heritability. Approximately, 2905 Holstein heifers (11–15 months of age) were used for circulating AMH measurements and then genotyped for SNP (single-nucleotide polymorphism) markers, and their pedigree data from the last four generations were collected. The genomic heritability was 0.36 ± 0.03 for AMH [94], and another research team also reported similar heritability (0.46 ± 0.31) by using data from 198 Canadian Holstein cows [95]. The heritability of AMH on pedigree-based information was estimated as 0.43 ± 0.07 [94]. These estimated heritability traits (genomic and pedigree) were higher than any previously published reports for traits related to reproduction in cattle [93]. The heritability of AFC in dairy heifers and cows was 0.25 ± 0.13 and 0.31 ± 0.14, respectively. As per previous reports, AFC was revealed to be a moderately heritable genetic trait [96].

By genome-wide association analysis, a relevant overlap was found between the genes influencing AMH concentrations and those which affect the superovulatory traits in cattle [94]. For example, prostaglandin-endoperoxide synthase 1 (PTSGS1) was associated with *AMH* [94], a gene that had been found positively correlated with the number of ova/collectable and viable embryos in cattle [97]. Given that numerous findings [55,94,95] have revealed that fertility might be improved by genetic selection of the size of the ovarian pool (as assessed by AFC and AMH) in cattle, the economically relevant production traits that correlate with potential positive genetic traits need to be determined as AMH was not associated with the level of milk production [79], while AFC was negatively correlated with genetic merit for milk fat concentration [96].

## 9. Factors Affecting AMH

### 9.1. Nutrition

The ovarian follicle pool is ascertained during fetal life [98]. Consequently, pregnant dam management in cattle has a pivotal impact on the environment of conceptus development, thus impacting on the establishment of the ovarian pool. Hypothetically, in the first trimester of a pregnant dam, restriction of dietary nutrition up to 60% of maternal requirement has a lifelong impact on the establishment of ovarian reserves in offspring (beef cattle) [99]. The restriction of nutrition has a significant impact on the ovarian reserves of heifers born to experimental mothers, reduced plasma AMH levels (four months to 1.8 years), lower AFC (seven weeks to 1.6 years), and increased FSH concentration [44,99]. Another study found that a high level of protein fed to cattle during gestation impaired the AFC in offspring [100], predicting that an imbalanced diet and other fetal life events could have a long-lasting impact on the later fertile life of cattle.

### 9.2. Hormones

Endocrine-disrupting chemicals (natural and artificial) have a potential impact on the physiology, offspring, and reproduction of animals [101]. In sheep, excessive testosterone exposure during pregnancy (especially during day 30–90) reduces the AMH expression in granulosa cells of preantral follicles. Moreover, AMH expression levels increase in young adult antral follicles as compared to controls, while no effect was detected in prepubescent lambs [30]. In Chinese goats (Chongming White), superovulation treatment with CIDR–FSH–PGF2a–LH increases the ovulation and level of FSH and AMH in serum [82]. The above reports revealed that testosterone and inhibin immunization before superovulation brings about changes in AMH expression and concentration, suggesting that these hormones have a role in the regulation of ovarian reserves.

### 9.3. Disease

Ovarian diseases can lead to altered ovarian function, contributing to the follicular persistence and endocrine/paracrine changes found in domestic animals. AMH emerged as a potential biomarker for several ovarian tumors in cows [56,102,103], mares [104,105], dogs [106], and cats [107]. Moreover, AMH has been reported as a potential tool for monitoring bovine granulosa–theca cell tumor (GTCT) spontaneous recovery [108]. In women, polycystic ovarian syndrome (PCOS) showed a three-fold higher AMH level as compared to normal, indicating the higher load of growing follicles [109], whereas in cattle, the expression and concentration of AMH are clearly altered in the course of follicular persistence and in developed cystic ovarian disease (COD) [110]. Furthermore, elevated AMH has been reported in bitches with luteinized follicular cysts [106]. As per the above reports, AMH can be used as potential diagnostic tool for ovarian tumors in many domestic animals, but the predictive ability for cystic diseases is still controversial, and so further studies are required.

Naturally, gestation and lactation traits are interlinked by hypothesis. Cows were selected as having a chronic mammary gland infection (increased somatic cell count (SCC)) [111], and daughter heifers have low circulating AMH levels [57]. In conclusion, severe/persistent mammary gland infections have an impact on the offspring ovarian reserves during gestation.

### 9.4. Granulosa Cells

In superstimulation treatment, cattle with low AFC show a poor response [45]. To test this phenomenon, an in vitro model was designed to check the response of the granulosa cells among low and high AFC individuals under different concentrations of FSH. The expression and quantity of AMH mRNA were measured. Overall results indicated that the abundance and expression of AMH were relatively lower in the low AFC group in comparison with the high AFC group. This report revealed that the low AFC granulosa cells responded less to FSH, while the FSH concentration in low AFC cattle was chronically high [44,45,48,50]. The debate over whether cattle with lower ovarian reserves show less of a refractory response to FSH during the reproductive cycle and superovulatory treatment is still unclear or only partially explained [45,49,83,84,85].

## 10. Conclusions

Recently, AMH has become a hot topic for researchers due to its ability to make predictions regarding the ovarian reserve pool. Due to its static nature, the collection of a single dose of blood has become more popular in the field of ART. In contrast to AFC, the strong genomic and pedigree-based heritability of AMH strengthens the case for its use as a reproductive biomarker of fertility. The peripheral AMH level has become representative of the ovarian reserve pool and is currently a promising marker for fertility and a diagnostic marker for ovarian disorders in domestic animals. The relationship with reproductive performance parameters—i.e., breed, age, longevity, fertility, and ovarian reserves (for heifers)—needs further confirmation within large herds. The impact of management factors (disease and nutrition), hormones, and lower response of AFC to cattle and goat granulosa cells opens a new pathway for future research. In many domestic farm animal species, the association of AMH to fertility superovulation and disease remains unexplored.

## Figures and Tables

**Figure 1 genes-10-01009-f001:**
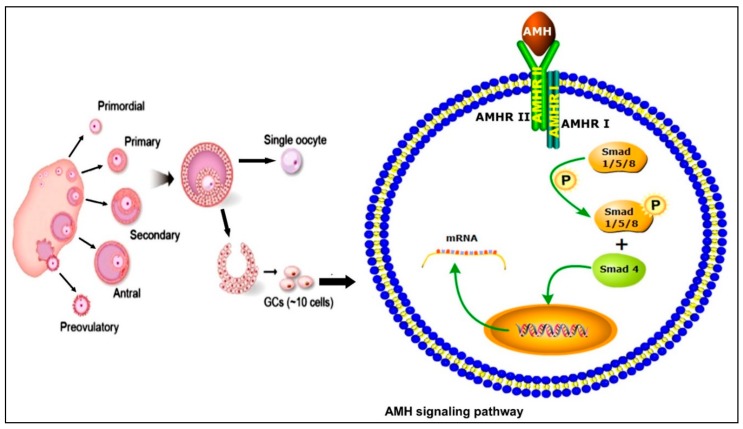
Schematic representation of different stages of ovarian follicular development and the anti-Müllerian hormone (AMH) signaling mechanism in granulosa cells (GCs). Upon ligand binding, the type II receptor activates the type I receptor which, in turn, activates the phosphorylation of Smads. These receptor-activated Smads interact with Smad4 and translocate to the nucleus to regulate gene transcription.

**Figure 2 genes-10-01009-f002:**
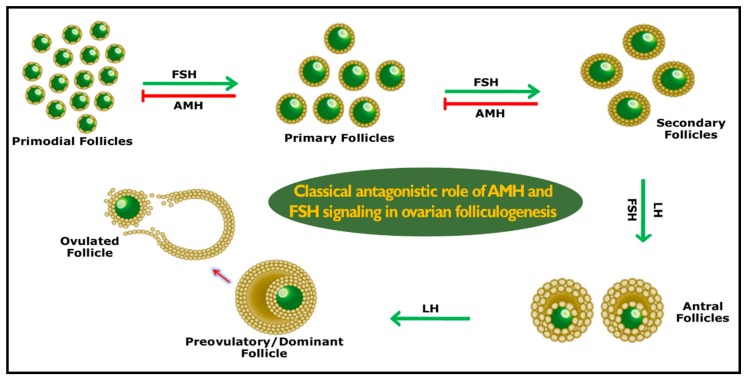
Schematic role of AMH which avoids the premature exhaustion of ovarian follicular reserves and selection of a dominant follicular wave. AMH works inversely to follicle-stimulating hormone (FSH) in accomplishing the aforementioned tasks. LH = luteinizing hormone.

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
