# Peer review of "AMH: Could It Be Used as A Biomarker for Fertility and Superovulation in Domestic Animals?"

_genes, 2019, doi:10.3390/genes10121009_

Round 1

Reviewer 1 Report

Major comments;

The current review is well written and covers a Hot topic. However, there are several reviews that cover the same subject [1-4]. Therefore, the reviewer has some suggestions which might improve the current review and make it more relevant;

1.       AMH also have been recently proposed as a potential biomarker for precocious puberty and weaning [5, 6]. It is well known that Heifers with precocious puberty (≤10 months) attain puberty and breed at less cost than those with later puberty [7]. Moreover, precocious puberty allows the heifer to have more estrous cycles before breeding age, increased first-service conception rate [8], earlier pregnancy [9], and enhanced lifetime productivity [10]. Therefore, it is strongly recommended to give a hint about that in the current review.

2.       None of the previously mentioned reviews [1-4] addressed the utility of AMH as a diagnostic biomarker for infertility; for example, AMH emerged as a potential biomarker for several ovarian tumors in cows [11-13], mares [14, 15], dogs [16] and cats [17]. Moreover, AMH has been reported as a potential tool for monitoring bovine GTCT spontaneous recovery [18]. Furthermore, elevated AMH has been reported in the bitch with luteinized follicular cysts [16]. On the other hand, AMH failed to be diagnostic biomarkers for cystic ovarian disease in cows [11]. Therefore, it is strongly recommended to add a section (one or two paragraphs) in the current review to address the potential diagnostic value of AMH for different ovarian and genital abnormalities in domestic animals.

3.       In Lines 120-125 & Lines 181-186; The fact that AMH is stable during the bovine estrous cycle is correct. However, the cited studies did not assay AMH on a daily basis during and/or spontaneous estrus cycle. For example, Rico et al. reported that the AMH profile was relatively stable over a 20-day investigation interspersed with two parenteral administrations of PGF2α spaced 11 days apart. Therefore, it is more relevant to add and cite the study which assayed AMH on a daily basis during the spontaneous estrous cycle in cows [11].

Minor comments;

Several Abbreviations have been mentioned without providing the full name at first time (e.g., MAPK…etc)

References

1.            Mossa F, Ireland JJ. Physiology and endocrinology symposium: Anti-Mullerian hormone: a biomarker for the ovarian reserve, ovarian function, and fertility in dairy cows. J Anim Sci 2019; 97:1446-1455.

2.            Mossa F, Jimenez-Krassel F, Scheetz D, Weber-Nielsen M, Evans ACO, Ireland JJ. Anti-Mullerian Hormone (AMH) and fertility management in agricultural species. Reproduction 2017; 154:R1-r11.

3.            Ireland JJ, Smith GW, Scheetz D, Jimenez-Krassel F, Folger JK, Ireland JLH, Mossa F, Lonergan P, Evans ACO. Does size matter in females? An overview of the impact of the high variation in the ovarian reserve on ovarian function and fertility, utility of anti-Müllerian hormone as a diagnostic marker for fertility and causes of variation in the ovarian reserve in cattle. Reproduction, Fertility and Development 2010; 23:1-14.

4.            Monniaux D, Drouilhet L, Rico C, Estienne A, Jarrier P, Touze JL, Sapa J, Phocas F, Dupont J, Dalbies-Tran R, Fabre S. Regulation of anti-Mullerian hormone production in domestic animals. Reprod Fertil Dev 2012; 25:1-16.

5.            El-Sheikh Ali H, Kitahara G, Takahashi T, Mido S, Sadawy M, Kobayashi I, Hemmi K, Osawa T. Plasma anti-Mullerian hormone profile in heifers from birth through puberty and relationship with puberty onset. Biol Reprod 2017; 97:153-161.

6.            Santa Cruz R, Cushman R, Viñoles C. Antral follicular count is a tool that may allow the selection of more precocious Bradford heifers at weaning. Theriogenology 2018; 119:35-42.

7.            Wehrman ME, Kojima FN, Sanchez T, Mariscal DV, Kinder JE. Incidence of precocious puberty in developing beef heifers. J Anim Sci 1996; 74:2462-2467.

8.            Buskirk DD, Faulkner DB, Ireland FA. Increased postweaning gain of beef heifers enhances fertility and milk production. J Anim Sci 1995; 73:937-946.

9.            Bagley CP. Nutritional management of replacement beef heifers: a review. J Anim Sci 1993; 71:3155-3163.

10.          Lesmeister J, Burfening P, Blackwell R. Date of first calving in beef cows and subsequent calf production. Journal of Animal Science 1973; 36:1-6.

11.          El-Sheikh Ali H, Kitahara G, Nibe K, Yamaguchi R, Horii Y, Zaabel S, Osawa T. Plasma anti-Müllerian hormone as a biomarker for bovine granulosa-theca cell tumors: Comparison with immunoreactive inhibin and ovarian steroid concentrations. Theriogenology 2013; 80:940-949.

12.          El-Sheikh Ali H, Kitahara G, Nibe K, Osawa T. Endocrinological characterization of an ovarian sex cord–stromal tumor with a Sertoli cell pattern in a Japanese Black cow. Reproduction in Domestic Animals; 0.

13.          Kitahara G, Nambo Y, El-Sheikh Ali H, Kajisa M, Tani M, Nibe K, Kamimura S. Anti-Müllerian Hormone Profiles as a Novel Biomarker to Diagnose Granulosa-theca Cell Tumors in Cattle. Journal of Reproduction and Development 2012; 58:98-104.

14.          BALL BA, ALMEIDA J, CONLEY AJ. Determination of serum anti-Müllerian hormone concentrations for the diagnosis of granulosa-cell tumours in mares. Equine Veterinary Journal 2013; 45:199-203.

15.          Almeida J, Ball BA, Conley AJ, Place NJ, Liu IKM, Scholtz EL, Mathewson L, Stanley SD, Moeller BC. Biological and clinical significance of anti-Müllerian hormone determination in blood serum of the mare. Theriogenology 2011; 76:1393-1403.

16.          Walter B, Coelfen A, Jager K, Reese S, Meyer-Lindenberg A, Aupperle-Lellbach H. Anti-Muellerian hormone concentration in bitches with histopathologically diagnosed ovarian tumours and cysts. Reprod Domest Anim 2018; 53:784-792.

17.          Heaps LA, Scudder CJ, Lipscomb VJ, Steinbach SM, Priestnall SL, Martineau H, Szladovits B, Fowkes RC, Garden OA. Serum anti-Müllerian hormone concentrations before and after treatment of an ovarian granulosa cell tumour in a cat. JFMS open reports 2017; 3:2055116917722701-2055116917722701.

18.          El-Sheikh Ali H, Kitahara G, Torisu S, Nibe K, Kaneko Y, Hidaka Y, Osawa T. Evidence of Spontaneous Recovery of Granulosa-Theca Cell Tumour in a Heifer: A Retrospective Report. Reproduction in Domestic Animals 2015; 50:696-703.

Author Response

First of all, we are very thankful to the reviewer for giving key suggestions and allowed us to improve the manuscript in a better way. The given suggestions are constructive for making the review story more attractive. We have revised the manuscript according to the comments and suggestions, and the amendments are highlighted with red and blue colors in the revised manuscript. Particularly, the rewriting part is highlighted in blue. Below you will find our point-by-point responses to your comments. The whole manuscript has been carefully rechecked. We do hope we could understand your questions correctly and have given the right answers in the revised manuscript. Please feel free to inform us if there are still some questions. Thank you very much in advance!

Your suggestions gave us much to learn and helped improve our scientific writing to a great extent.!

Response to Reviewer 1:

Comment 1: We are very thankful for constructive inputs. we have incorporated a section regarding precocious puberty from line 179 to 190.

Comment 2: Thank you for your critical review. We have included fair literature reports related to “AMH as diagnostic marker”, under a separate paragraph in Disease sub-section. You can refer to line 281 to 292. We have also incorporated the respective points in the abstract and conclusion part of this manuscript.

Comment 3: As per the suggestion of the worthy reviewer, we are pleased to add all the relevant citations in the manuscript.

Comment 4: Abbreviation problems have been addressed throughout the manuscript.

Reviewer 2 Report

In many places I was unable to understand what the authors meant because of poor use of English. This MS needs very extensive editing by an editor fluent in English before it will be fit for publication. This is not just a question of grammar or spelling, but one where the English expression is ambiguous, misleading or nonsensical. For example, the main conclusion in the Abstract says "... we concluded that due to the static nature of AMH, it may act as a potential predictor of fertility ...". If AMH levels in the blood (is this what they are referring to) are "static", ie unvarying, how can they be a predictor of anything.

The manuscript is also littered with other errors. On line 40 there is a reference to "KDa" but surely in the standard nomenclature the symbol for kilo is a lower case "k" not an upper case "K". IN the caption for figure 1, several spaces are missing between words (perhaps this is an issue of the PDF conversion, but I suspect not).

Examples of obscure wording include Lines 48 -50: "... AMH was an indirect member of the TGF-ß superfamily but due to the signalling mechanism of BMPs, it became a permanent member of this family ...". I am not sure what this means. Perhaps it relates to the history of discovery and clarification of AMH's biochemistry (words like "was" and "became" suggest this), but if this is the case, is it relevant in a section entitled "Regulation of AMH"? Indeed this section seems more involved in the mechanism of action of AMH than in the regulation of AMH, so perhaps the section title needs changing to, perhaps, something like "AMH signalling pathways".

Problems such as this will necessitate a complete rewrite of the text body.

There are also problems in the reference list. For example Ref 56 is incompletely/incorrectly cited. See PMID: 21366975.

The figures also need attention. For example in figure 1 the blocks with SMADs downstream of AMHR are labelled with minute text that is virtually unintelligible unless the page size is expanded to well above 100% on screen. There is no reason why the lettering cannot be made clearer throughout. The figure caption describes the rightmost part of the figure and ignores the left and centre parts unexplained. Figure 2 has a caption that is perfunctory. Checking the body text in the home of clarification, I note at line 99 "... reduces the sensitivity of parental and small antral follicles to FSH while modulating follicular development ...".  Parental folicles???

This manuscript needs thorough rewriting to ensure that is it written in clear and unambiguous English, and that it is free from errors, before it is suitable for resubmission. It is unfortunate that the authors did not do this before the original submission, which would have saved editors and reviewers their valuable time.

Author Response

First of all, we are very thankful to the reviewer for giving key suggestions and allowed us to improve the manuscript in a better way. The given suggestions are very helpful for making the review story more attractive. We have revised the manuscript according to the comments and suggestions, and the amendments are highlighted with red and blue colors in the revised manuscript. Particularly, the rewriting part is highlighted in blue. Below you will find our point-by-point responses to your comments. The whole manuscript has been carefully rechecked. We do hope we could understand your questions correctly and have given the right answers in the revised manuscript. Please feel free to inform us if there are still some questions. Thank you very much in advance!

Your suggestions gave us much to learn and helped improve our scientific and English writing to a great extent!

Response to Reviewer 2:

Comment 1: We are very thankful for constructive suggestions regarding English quality. An English expert has reviewed the manuscript, and given suggestions are being incorporated.

Comment 2: We have sufficiently addressed these comments. We have addressed the abbreviations and units’ issues throughout the manuscripts.

Comment 3: As per suggestion of the worthy reviewer, we are pleased to address the “phrase problems”. We have changed the subheading according to the reviewer suggestion (line 48). We are very much sorry for improper citation, and we have corrected that citation (previously 56, now 57).

Comment 4: we are very much pleased to incorporate remodified figure 1, we hope that now the figure quality and figure legend caption are reasonable. We are sorry for misspelling the “preantral” in line 99. these types of mistakes are extensively checked and corrected throughout the manuscript. 

Round 2

Reviewer 1 Report

Some references does not have Issue, Volume and Pages 

Author Response

First of all, we are very thankful to the reviewer for giving key suggestions and allowed us to improve the manuscript in a better way. The given suggestions are constructive for making the review story more attractive. We have revised the manuscript according to the comments and suggestions, and the amendments are highlighted with orange and purple colors in the revised manuscript. Particularly, the rewriting part is highlighted in red. The whole manuscript has been carefully rechecked. We do hope we could understand your questions correctly and have given the right answers in the revised manuscript. Please feel free to inform us if there are still some questions. Thank you very much in advance!

Yours sincerely,

Corresponding Authors:  Prof. Dr. Huabin Zhu

Response to Reviewer 1:

Comment 1: We are very thankful to you for accepting our revised inputs. As per the suggestion of worthy reviewer, we have carefully checked all the bibliography and incorporate the missing information of citations. Once again, many thanks for critical review and constructive inputs.

Reviewer 2 Report

Alas the second version of the manuscript is almost as incomprehensible as the first version. If the journal decides to pursue this manuscript for publication, it will need to be thoroughly rewritten by a competent science editor who is fluent in English.

Working through this revision, I compiled a whole page of comments just on the abstract. I don't feel it is worth more of my time to work my way laboriously through the remainder, particularly since the conclusion seems to be that more research is needed to establish if AMH has any value as a biomarker for fertility etc...

Line No

Comment

16

“… as compare to high concentration afterward” need to clearly what it is that this is afterwards of, otherwise this statement provides no useful information.

16

as compared

16

AFC --- define abbreviation

16

lower pregnancy rate

20

“ before, during and post estrus cycle;”  I am not sure what “before estrous cycle” means – pre puberty? post-estrous cycle ? does this mean anestrus? It is the nature of cycles that they go on and on – they don’t have beginings and endings in the same way that a line has a beginning and end but a circle does not.

“estrus cycle”  -  estrus is a noun. The correct form here is the adjective ie, estrous.

23

Many factors disrupt the circulatory levels of AMH in 23 plasma plasma.

Is “disrupt” the correct term? would “affect” be better. A comma is needed after “plasma”.

“disrupt” might be appropriate if all the examples were qualified as abnormal states, eg like poor nutrition …

“endocrine changes” is meaningless – the body is awash with hormones that change in response to circadian rhythms etc (and changes in AMH are also an endocrine change, making this a partly circular argument.

Likewise “activity of granulosa cells” what activity in particular? Production of AMH is an activity of granulosa cells – if this is the activity in question, then there is no useful information conveyed here.

24-26

“ we concluded that due to the relatively static nature of AMH (independent of individual differences among domestic animal species), it may act as a potential predictor of fertility …”

The wording here is still obscure. I am guessing the following might be what the authors intend to convey:

… since plasma AMH concentration does not vary  across the estrous cycle within individuals, but there are significant variations between individuals, that these between individual variations might reflect underlying differences in ovarian function so plasma AMH levels may be predictors of …

However if that is the main conclusion I am not sure how this differs from a number of previous publications.

27

“ However, due to limited research work related to domestic animals, this potential of AMH is still mysterious”

A strange wording. However, the conclusion is that more research is needed to establish if plasma AMH concentration is a predictor for fertility, effectiveness of superovulation treatments, or ovarian disorders. In other words the authors at this stage state that they have nothing new to add.

Author Response

First of all, we are very thankful to the reviewer for giving key suggestions and allowed us to improve the manuscript in a better way. The given suggestions are constructive for making the review story more attractive. We have revised the manuscript according to the comments and suggestions, and the amendments are highlighted with orange and purple colors in the revised manuscript. Particularly, the rewriting part is highlighted in red. The whole manuscript has been carefully rechecked. We do hope we could understand your questions correctly and have given the right answers in the revised manuscript. Please feel free to inform us if there are still some questions. Thank you very much in advance!

Yours sincerely,

Corresponding Authors:  Prof. Dr. Huabin Zhu

Response to Reviewer 2:

Comment 1: We are very thankful for constructive suggestions regarding English quality. Technical English editing has been carried out by MDPI special English editing services. MDPI English editing certificate attached below. All of the reviewer's suggestions have been incorporated.

Comment 2: Thank you very much. We have modified abstract according to your suggestions: Changes are highlighted with red color. Thank you for understanding.
